# Screening and Identification of Novel Soluble Epoxide Hydrolase Inhibitors from Corn Gluten Peptides

**DOI:** 10.3390/foods11223695

**Published:** 2022-11-18

**Authors:** Jiamin Dang, Shuangkui Du, Liying Wang

**Affiliations:** 1College of Food Science and Engineering, Northwest A&F University, Xianyang 712100, China; 2Engineering Research Center of Grain and Oil Functionalized Processing, Universities of Shaanxi Province, Xianyang 712100, China

**Keywords:** corn gluten, peptide, soluble epoxide hydrolase, inhibitor, molecular docking

## Abstract

The objective of this study was to investigate the soluble epoxide hydrolase (sEH) inhibitory properties of corn gluten peptides. In total, 400 dipeptides and 8000 tripeptides were first virtually screened by molecular docking and 30 potential sEH inhibitory peptides were selected. Among them, WEY, WWY, WYW, YFW, and YFY showed the highest sEH inhibitory activities with IC_50_ values of 55.41 ± 1.55, 68.80 ± 7.72, 70.66 ± 9.90, 96.00 ± 7.5, and 94.06 ± 12.86 μM, respectively. These five peptides all behaved as mixed-type inhibitors and were predicted to form hydrogen bond interactions mainly with Asp333, a key residue located in the catalytic active site of sEH. Moreover, it was found that the corn gluten hydrolysates of Alcalase, Flavourzyme, pepsin and pancreatin all exhibited high sEH inhibitory activities, with IC_50_ values of 1.07 ± 0.08, 1.19 ± 0.24, and 1.46 ± 0.31 mg/mL, respectively. In addition, the sEH inhibitory peptides WYW, YFW, and YFY were successfully identified from the corn gluten hydrolysates by Alcalase using nano-LC-MS/MS. This study demonstrated the sEH inhibitory capacity of peptides for the first time and corn gluten might be a promising food protein source for discovering novel natural sEH inhibitory peptides.

## 1. Introduction

Soluble epoxide hydrolase (sEH) is a member of the α/β-hydrolase family and is widely expressed in tissues, such as the liver, kidney, heart, lung, brain, and intestine [1]. It is a homodimer and each monomer comprises a C-terminal hydrolase and an N-terminal phosphatase. The C-terminal hydrolase predominantly degrades epoxy-fatty acids, for example, epoxyeicosatrienoic acids (EETs), to their corresponding diols [2]. EETs are known to be generated from arachidonic acid via the cytochrome P450 epoxygenase pathway and have been shown to have anti-inflammatory, hypotensive, hypoglycemic, and neuroprotective properties. However, sEH can convert the active EETs to inactive dihydroxyeicosatrienoic acids (DHETs), causing them to lose their beneficial effects [3]. Therefore, sEH has become a new potential target for many metabolic, inflammatory, vascular, and neurodegenerative diseases [4]. Inhibiting the activity of sEH can effectively prevent the hydrolysis of EETs and increase the level of intracellular EETs so they can exert their physiological functions [5].

Discovering novel sEH inhibitors through chemical synthesis or from natural products has attracted a lot of attention. It has been found that most of the sEH inhibitors mainly include urea and amide groups [6]. Compared with synthesized drugs, natural inhibitors are considered safer and are thought to have fewer adverse effects. As a consequence, the interest in screening effective sEH inhibitors from natural sources has increased in the last decade [7]. Until now, many sEH inhibitors have been identified from traditional Chinese herbs, such as *Alisma orientale* [8], *Scutellaria baicalensis* [9], *Leonurus japonicus* [10], and *Eucommia ulmoides* [11]. However, these sEH inhibitors often have poor water solubility, poor intestinal absorption, and low bioavailability, resulting in undesirable pharmacokinetic properties and a decrease in potency after oral intake [12]. All this implies the importance of discovering possible sEH inhibitors from common nutritional components of foods.

Bioactive peptides prepared from food proteins by enzymatic hydrolysis or microbial fermentation have been considered promising nutraceutical treasures due to their excellent health-promoting benefits [13]. Corn gluten meal is a good food protein source for bioactive peptides and peptides generated from corn gluten have been demonstrated to have various biological activities, such as antioxidant, anti-inflammatory, antihypertensive, antihyperglycemic, hepatoprotective, alcohol metabolism-facilitating, anticancer, and antimicrobial activities [14]. In our previous study, it was found that peptides generated from corn gluten hydrolysis by the enzyme Alcalase had many free radicals and high intracellular reactive oxygen species scavenging capacity [15,16]. Moreover, corn gluten peptides could significantly suppress the expression of Epoxide Hydrolase 2 (EPHX2), the human gene encoding sEH, in hydrogen peroxide-induced HepG2 cells [17]. However, the effects of corn gluten peptides on sEH activity have never been reported.

Molecular docking has been widely used in the rapid screening of bioactive compounds by stimulating interactions between target proteins and candidate compounds. For example, Zhao et al. screened potential inhibitors against the main protease of SARS-CoV-2 from lactoferrin-derived peptides using molecular docking [18]. Yu et al. also successfully identified xanthine oxidase inhibitors from egg white-derived peptides using molecular docking [19]. Compared with traditional methods, the molecular docking-based in silico approach is highly efficient, time-saving, and low-cost. Therefore, in this study, the potential sEH inhibitory capacity of peptides was first explored through the computational docking of peptides into the active site of sEH. The peptides with a strong binding ability to sEH were selected and their sEH inhibitory activities were determined, as well as their inhibition modes and interactions with sEH. Furthermore, the inhibitory properties of corn gluten peptides from different proteases on sEH activity were studied and the corn gluten peptides were identified by nano-LC-MS/MS.

## 2. Materials and Methods

### 2.1. Materials and Reagents

Corn gluten meal was purchased from Xi’an Guowei Starch Co., Ltd. (Xi’an, Shaanxi, China). All peptides were synthesized by solid-phase procedures with a purity of over 95% by China Peptides Co., Ltd. (Shanghai, China). Bovine serum albumin (BSA), bis-Tris-HCl, dimethyl sulfoxide (DMSO), and methanol were purchased from Shanghai Aladdin Biochemical Technology Co., Ltd. (Shanghai, China). Human soluble epoxide hydrolase (sEH), 3-Phenyl-cyano(6-methoxy-2-naphthalenyl)methylester-2-oxirane acetic acid (PHOME), and 12-(3-adamantan-1-yl-ureido) dodecanoic acid (AUDA) were purchased from Cayman Chemical (Ann Arbor, MI, USA). Alcalase (1000 U/mg), papain (1000 U/mg), Flavourzyme (1000 U/mg), and Neutrase proteases (1000 U/mg) were purchased from Nanning Pangbo Bioengineering Co., Ltd. (Nanning, Guangxi, China). Pepsin (from porcine gastric mucosa, 3200 U/mg) and pancreatic (from porcine pancreas, 3×USP specifications) were purchased from Sigma-Aldrich (St. Louis, MO, USA). All other chemicals were of analytical grade.

### 2.2. Preparation of Corn Gluten Peptides

Corn gluten peptides were prepared through enzymatic hydrolysis according to the method by You et al. with minor modifications [20]. The corn gluten meal was dissolved in deionized water to obtain the aqueous solution of corn gluten (5%) and heated at 90 °C for 10 min to denature the protein. After cooling down (Alcalase: 50 °C; Flavourzyme: 50 °C; papain: 55 °C; Neutrase proteases: 45 °C), the pH was adjusted with 1M NaOH or HCl (Alcalase: 8; Flavourzyme: 7; Papain: 7; Neutrase proteases: 7) followed by addition of different enzymes at an enzyme: substrate of 4%. The corn gluten was hydrolyzed for 180 min in a thermally controlled incubator under constant stirring. In addition, an in vitro simulated gastrointestinal digestion experiment of corn gluten was carried out. The corn gluten (2%) was first hydrolyzed by pepsin at pH 2 and 37 °C for 90 min and then by pancreatin at pH 7 and 37 °C for 240 min. Both pepsin and pancreatin were added at an enzyme: substrate of 2%. At the end of hydrolysis, all the resulting mixtures were heated at 90 °C for 10 min to inactivate the enzymes.

Then, all the corn gluten hydrolysates were centrifuged at 4000 r/min for 10 min. The supernatant was fractionated using ultrafiltration with a molecular weight cut-off membrane of 1 kDa (Shanghai Laungy Membrane Separation Equipment Engineering Co., Ltd., Shanghai, China). The corn gluten hydrolysate fractions with a molecular weight less than 1 kDa were selected, lyophilized, and stored at −20 °C.

### 2.3. Virtual Screening of Potential sEH Inhibitory Peptides

Molecular docking was performed to screen potential sEH inhibitory peptides. The X-ray structure of human sEH (PDB code: 1ZD3) was used as the receptor [21]. The structures of the native ligand (4-{[(cyclohexylamino)carbonyl]amino}butanoic acid, NC4) in the X-ray structure of human sEH and 400 dipeptides and 8000 tripeptides were defined as ligands in Molecular Operating Environment (MOE) (Chemical Computing Group Inc., Montreal, QC, Canada) after energy minimization. The binding site of sEH was set with the coordinates: x = −17.400, y = −7.960, and z = 66.430. The 2D structures of peptides were generated by the Protein Builder module in MOE. Three docking tools, MOE, AutoDock Vina, and CDOCKER were used to verify the better suitable docking method of redocking the native ligand NC4 to the binding site of sEH. The position of the native ligand NC4 in the X-ray structure of sEH was defined as the binding site.

For MOE, before docking, the force field of AMBER10: EHT and the implicit solvation model of the Reaction Field (R-field) were selected. The 2D structures of peptides were converted to 3D structures through an energy minimization module under the force field of AMBER10: EHT in MOE. The docking workflow followed the “induced fit” protocol, in which the side chains of the receptor pocket were allowed to move according to ligand conformations, with a constraint on their positions. The weight used for tethering side chain atoms to their original positions was 10. For each ligand, all docked poses of which were ranked by London dG scoring first, and then a force field refinement was carried out on the top 30 poses followed by a rescoring of GBVI/WSA dG. The conformation with the lowest binding free energy was finally identified as the best probable binding mode.

For AutoDock Vina, the X-ray structure of sEH was converted to a PDBQT file that contains a protein structure with hydrogens in all polar residues in AutoDock4. Additionally, the ligands were also converted to a PDBQT file and all the bonds were set to be rotatable in AutoDock4. All bonds of ligands were set to be rotatable. Lamarckian Genetic Algorithm (LGA) is used to calculate the flexible docking of protein-fixed ligands. AutoDock Vina Extended automatically calculates the grid maps and clusters the results in a way transparent to the user. The best conformation was chosen with the lowest binding energy after the docking search was completed.

For CDOCKER, the water molecules and the native ligands were removed from the complex crystal structure and then prepared by Prepare Protein module in Discovery Studio (DS). The 2D structures of the ligands were prepared (including adding hydrogen atoms, ionizing at the pH range from 6.5 to 8.5, generating tautomers and isomers, fixing bad valencies, and generating 3D coordinates) by Prepare Ligands module in DS. The peptides were prepared based on their sequence, and then modification and energy minimization were performed by the Energy Minimize procedure. Finally, the Prepare Ligands module was carried out to make the peptides ready before docking. The CDOCKER protocol was conducted between the ligands and the receptor. The conformation with the lowest CDOCKER Energy was finally recognized as the most stable conformation.

After the molecular docking, the interactions between the peptides and sEH including the hydrogen bonds and the bond lengths were analyzed using PyMol [22].

### 2.4. Validation of the Reaction Conditions between the sEH and Substrate

The effects of the concentrations of the enzyme, substrate, and the hydrolysis time of sEH on PHOME were determined according to the methods reported previously [11,23]. Briefly, 130 µL of 12.5 or 62.5 ng/mL human sEH solution (final concentration, dissolved in 25 mM bis-Tris-HCl containing 0.1% BSA, pH 7.0) and 20 µL of 10% methanol were added to a 96-well plate and incubated at room temperature for 15 min. Then, 50 µL of PHOME solutions at final concentrations of 0, 2.5, 5, 7.5, 10, 12.5, 15, 17.5, 20, and 22.5 µM (dissolved in 25 mM bis-Tris-HCl containing 0.1% BSA, pH 7.0) was added to initiate the reaction. After incubating at 37 °C for 5 to 50 min, the fluorescence intensity was measured at an emission wavelength of 465 nm and an excitation wavelength of 330 nm. Bis-Tris-HCl instead of PHOME was used as the blank control. The reaction rate of the system was calculated as follows:(1)$$\mathrm{The}\;\mathrm{reaction}\;\mathrm{rate}\;(\mathrm{RFU}/\min)=\frac{S_{t}\;{-\;C}_{t}}{t}$$
where S_t_ (RFU) and C_t_ (RFU) represent the fluorescence intensity of the sample and blank group at t min, respectively. The t (min) represents the reaction time.

### 2.5. Determination of the sEH Inhibitory Activities of Peptides

The sEH inhibitory activities of peptides were determined as above described. Briefly, 130 µL of 62.5 ng/mL human sEH solution (final concentrations, dissolved in 25 mM bis-Tris-HCl containing 0.1% BSA, pH 7.0) and 20 µL of peptides (final concentrations: synthesized peptides, 20, 40, 60, 80, and 100 µM; corn gluten peptides, 1, 2, 3, 4, and 5 mg/mL, respectively) were added to 96-well plate followed by adding 50 µL of 10 µM PHOME solution (final concentration, dissolved in 25 mM bis-Tris-HCl containing 0.1% BSA, pH 7.0). After incubating at 37 °C for 5 to 40 min, the fluorescence intensity of the mixture at an emission wavelength of 465 nm and an excitation wavelength of 330 nm was determined. AUDA and bis-Tris-HCl instead of peptide samples were used as the positive and blank control, respectively. The sEH inhibitory activity was calculated as follows:(2)$$\mathrm{sEH}\;\mathrm{inhibitory}\;\mathrm{activity}\;(\%)=\frac{C_{40}\;{-\;S}_{40}}{C_{40}\;{-\;C}_{0}}\text{×}100\%$$
where C_40_ (RFU) and C_0_ (RFU) represented the fluorescence intensity of the blank control group at 40 and 0 min, respectively. S_40_ (RFU) represented the fluorescence intensity of the sample group at 40 min.

The median inhibitory concentration (IC_50_)—the concentration inhibiting 50% of sEH enzyme activity was calculated by through linear regression according to the percent inhibitory activity versus the sample concentration curve.

The sEH inhibition mode of peptides was determined using the Lineweaver−Burk double-reciprocal plot with PHOME and peptide concentrations ranging from 1 to 30 µM and 20 to 200 µM (final concentration), respectively.

### 2.6. Identification of Corn Gluten Peptides Using Nano LC-MS/MS

The peptide sequences of corn gluten hydrolysate were identified using nano-LC-MS/MS. Briefly, 5 μL of the sample was first loaded onto a nanocolumn (Acclaim PepMap RPLC C18, 150 μm × 150 mm, 3 μm, 100 Å; Dr. Maisch GmbH, Ammerbuch, Germany) in an Easy-nLC 1200 system (Thermo Fisher Scientific, Waltham, MA, USA). The sample was eluted by a linear gradient with mobile phase B (20% 0.1% formic acid in water—80% acetonitrile) from 4% to 8% for 2 min, from 8% to 28% for 43 min, from 28% to 40% for 10 min, from 40% to 95% for 1 min, and from 95% to 95% for 10 min. The mobile phase A was 0.1% formic acid in water. The flow rate was set at 600 nL/min.

Then, the sample was analyzed using Q Exactive™ Hybrid Quadrupole-Orbitrap™ Mass Spectrometer (Thermo Fisher Scientific). The electrospray voltage was set at 2.2 kV and the capillary temperature was set at 270 °C. The m/z for the full scan was 300 to 1000. The activation type was selected as HCD and the normalized collision energy was set at 28. Up to the top 20, the most intense peptide ions from the preview scan in the Orbitrap were used for the data-dependent acquisition. The raw MS file was analyzed and searched against the UniProt *Zea mays* L. (corn) database using Byonic.

### 2.7. Physicochemical Property and Toxicity Analysis

In addition, the sEH inhibitory peptides were analyzed for physicochemical properties and toxicity. The physicochemical properties of the identified peptides, including isoelectric point (pI), net charge at neutral pH, and water solubility, were analyzed by the peptide property calculator, which is available at http://www.innovagen.com/proteomics (accessed on 10 October 2022). The prediction of peptide toxicity using ToxinPred is available at https://webs.iiitd.edu.in/raghava/toxinpred/multisubmit.php (accessed on 10 October 2022).

### 2.8. Statistical Analysis

All data were presented as mean ± SD (*n* = 3). The difference in significance between the two groups was analyzed by Student’s t-test, and that between three and more groups was analyzed by one-way analysis of variance (ANOVA) followed by the Duncan multiple comparisons. Statistical analysis was performed by SPSS 19.0 with the significance level at *p* < 0.05.

## 3. Results and Discussion

### 3.1. The Optimal Reaction Conditions between the sEH and Substrate

PHOME is widely used in the high-throughput screening of sEH inhibitors. As a substrate, PHOME is highly sensitive, hydrolytically stable, and shows a large change in the fluorescence in the hydrolysis reaction [24]. When hydrolyzed by sEH, the non-fluorescent PHOME was converted to fluorescent 6-methoxy-2-naphthaldehyde. However, the inhibitor can competitively bind to the active site of sEH and lead to a decrease in the formation of the fluorescent products [25].

In the present study, the reaction conditions, including sEH and PHOME concentrations and reaction time, were optimized. A high reaction rate was observed when the sEH enzyme concentration of 62.5 ng/mL was used. The reaction rate increased significantly with the substrate concentration from 0 to 10 µM and decreased from 12.5 to 22.5 µM (Figure 1A). In addition, the reaction rate increased from 5 to 40 min and was maintained from 40 to 50 min with substrate concentrations ranging from 7.5 to 17.5 µM (Figure 1B). Therefore, sEH and PHOME concentrations at 62.5 ng/mL and 10 µM, respectively, and the reaction time of 40 min were selected for further determination of the sEH inhibitory activity.

### 3.2. Virtual Screening of sEH Inhibitory Peptides by Molecular Docking

In the past few years, the speed and accuracy of structural predictions and docking models have been significantly improved. Therefore, in this study, we carried out molecular docking of the sEH enzyme to screen potential sEH inhibitory peptides. It is known that the active side of sEH is located in a cavity with two narrow openings [6]. Therefore, in the present study, 400 dipeptides and 8000 tripeptides were selected and docked into the active site of sEH. Before the screening, three different docking tools—MOE, AutoDock Vina, and CDOCKER were validated by redocking the native ligand NC4 into the binding site of sEH. The root means square deviation (RMSD) between the redocked pose and the native pose in MOE, AutoDock Vina, and CDOCKER were calculated as 1.062, 1.939, and 2.445 Å, respectively. The lower RMSD means better redock performance. It was found that MOE showed the lowest RMSD, indicating that it had a good ability to reproduce the native conformation. The alignment between the docked pose and the native pose was as shown in Appendix A.

The 400 dipeptides and 8000 tripeptides were docked to sEH by MOE. After being docked, all peptides were ranked according to the score, namely, the binding free energy. The peptide–sEH complex with the lowest binding free energy was recognized as the most stable conformation and this peptide might have a strong binding ability to the active site of sEH. Finally, the top 20 tripeptides and the top 10 dipeptides were selected. It was noted that the tripeptides generally showed a lower binding free energy than dipeptides (Table 1). These 30 peptides were synthesized and their sEH inhibitory activities were determined. It was interesting to find that, at the final concentration of 100 μM, all the peptides showed sEH inhibitory activities ranging from 19.63% to 70.01% (Table 1). Among these peptides, WEY had the highest sEH inhibitory activity of 70.01% ± 6.25% followed by WYW, YFW, WWY, and YFY with 60.30% ± 5.60%, 57.38% ± 7.62%, 55.15%± 2.38%, and 52.96% ± 3.09%, respectively (*p* < 0.05). In addition, YMW, YRW, RV, FRY, WF, WLR, and YHY also exhibited sEH inhibitory activities of more than 40% at 100 µM. All this suggests that the small peptides might have potential as sEH inhibitors.

Recently, a number of natural phytochemicals have been demonstrated to display sEH inhibitory activities and their IC_50_ values mainly range from 1 to 100 μM. For instance, quercetin and kaempferol showed sEH inhibitory activities with IC_50_ values of 22.5 ± 0.9 and 31.3 ± 2.6 μM, respectively [11]. In addition, quercetin-3-O-arabinoside, ursolic acid, corosolic acid, and 2-oxopomolic acid were found to exhibit sEH inhibitory activities with IC_50_ values of 39.3 ± 3.4, 84.5 ± 9.5, 51.3 ± 4.9, and 11.4 ± 2.7 μM, respectively [26].

Many sEH inhibitors have urea and amide groups [27]. Some natural compounds containing urea and amide were discovered to have excellent sEH inhibitory activities. For example, several urea derivatives from Brassicales, such as maca (*Lepidium meyenii*), papaya (*Carica papaya*), and watercress (*Nasturtium officinale*), have been isolated and shown to have sEH inhibitory activities [28]. The amide-containing capsaicin and dihydrocapsaicin isolated from Capsicum chinense were reported to show sEH inhibitory activities with IC_50_ values of 5.6 ± 1.2 and 7.3 ± 0.7 μM, respectively [29]. In addition, the amide derivatives pellitorine, piperlonguminine, dihydropiperlonguminine, and flutamide from Scutellaria baicalensis have also been shown to have sEH inhibitory activities with IC_50_ values of 93.68 ± 4.52, 6.06 ± 0.12, 7.83 ± 0.52, and 6.32 ± 0.31 μM, respectively [9].

In our present study, it was found that the peptides WEY, WYW, YFW, WWY, and YFY all exhibited sEH inhibitory activities with a significant dose-effect from 20 to 100 μM (final concentration) (*p* < 0.05), as shown in Figure 2. Peptide WEY showed the lowest IC_50_ value of 55.41 ± 1.55 μM, followed by WWY and WYW of 68.80 ± 7.72 and 70.66 ± 9.90 μM, and then YFW and YFY of 96.00 ± 7.5 and 94.06 ± 12.86 μM, respectively (*p* < 0.05) (Table 1). While the IC_50_ value of the positive control of AUDA was determined to be 63.53 ± 9.03 nM in this study. It is known that the peptide bond in peptides is one of the amide groups. In addition, the tripeptides and dipeptides have small linear structures which may be appropriate for the narrow cavity of the active site of sEH. All this may contribute to the sEH inhibitory activity of peptides. However, what should be noted is that the sEH inhibitory activity of peptides was determined for the first time in our present study. This implies the potential of peptides as natural sEH inhibitors.

### 3.3. Inhibition Mode of sEH Inhibitory Peptides

The inhibition kinetics of peptides WYW, WWY, YFW, YFY, and WEY on sEH were determined. The reaction rates of the five peptides all increased and then remained at a high level with substrate PHOME ranging from 1 to 30 μM (Appendix A). To determine the inhibition mode of peptides, the Lineweaver−Burk double-reciprocal plots were conducted and the Michaelis constant (K_m_) and the maximum rate (V_max_) were calculated. It was found that, as shown in Figure 3, with the increase in peptide concentrations, the Km increased while the Vmax decreased, indicating that the five peptides all acted as competitive and non-competitive mixed inhibitors. What should be noted is that the adequacy of the Lineweaver−Burk double-reciprocal plots in this study may be due to the low binding affinity of peptides to sEH.

### 3.4. Interactions between Peptides and sEH Enzyme

To further understand the inhibition mechanisms of peptides on sEH, the intermolecular interactions between peptides and the sEH enzyme were analyzed. It is well-known that the hydrolase catalytic pocket of sEH is primarily composed of a catalytic triad Asp333-Asp495-His523 and two tyrosine residues, Tyr381 and Tyr465. When an epoxide enters the active site, the oxygen atom of the epoxide group will be recognized by Tyr381 and Tyr465, which in turn activate the epoxide ring-opening by Asp333 [7]. Asp333 is the key residue responsible for the hydrolase activity. The NH of the urea or amide group of many sEH inhibitors can act as a hydrogen bond donor and form a hydrogen bond with the oxygen atom of Asp333. Therefore, various urea and amide-containing compounds have been shown to be as competitive inhibitors for sEH [6].

In the present study, it was found that the peptides WEY, WYW, YFW, WWY, and YFY were predicted to form multiple interactions mainly including hydrogen bonds and π stackings with the residues of the active site of sEH, as shown in Figure 4. The information on the interactions between peptides and the sEH enzyme is summarized in Table 2. The hydrogen atom of NH_3_^+^ of the N-terminal amino acid residue of WEY, WYW, and YFW were all predicted to behave as hydrogen bond donors to form hydrogen bonds with the oxygen atom of Asp333. Two hydrogen bonds between the oxygen atom of Asp333 and the hydrogen atoms of the carbon backbone of the N-terminal amino acid residue and the NH of the middle amino acid residue of YFY were also observed. In addition, the peptides were predicted to form hydrogen bonds with other residues that belonged to the active site of sEH. It was noted that hydrogen bonds between WWY and the residues of Ser413 and Met418 not Asp333 of sEH were predicted. The hydrogen bonds between peptides and sEH all had satisfactory bond distances [30]. Moreover, various π stacking interactions, such as arene–cation stacking, arene–arene stacking, and arene–H stacking between peptides and the residues of sEH were formed. All of this might contribute to the binding and inhibition of peptides on sEH.

The amide group of chemically synthesized sEH inhibitors can form more hydrogen bond interactions with the key residues of the active site of sEH. For example, it was found that not only the NH of the amide group formed a hydrogen bond with the carboxyl of Asp333 (Asp335) but also the CO of the amide group formed hydrogen bonds with the hydroxy of Tyr381 (Tyr383) and Tyr465 (Tyr 466) of sEH [31,32]. In the present study, only a hydrogen bond between the NH of the amide group of YFY and the carboxyl of Asp333 of sEH was observed. For WEY, WYW, and YFW, it was the free NH_3_^+^ not the NH of the amide group that formed hydrogen bond interactions with Asp333. In addition, different from the amide-based synthesized compounds, no hydrogen bond interactions between peptides Tyr381 and Tyr465 were observed. This might be associated with the relatively low sEH inhibitory activity of peptides than the amide-based synthetic inhibitors. Generally, the structure-activity relationships among these peptides can be used to further refine structural models of the sEH improving the power of future computational approaches.

### 3.5. Physicochemical Property and Toxicity Analysis of sEH Inhibitory Peptides

The physicochemical properties of the identified peptides were analyzed by the peptide property calculator (Appendix A). The peptide WEY had a negative charge at pH 7 and good solubility. The other peptides showed poor water solubility and electroneutrality. All peptides were predicted without toxicity. These results provided the basis for further analysis of the pharmacokinetic properties including the absorption in the intestine, bioavailability, and half-time in animals and humans in the future.

### 3.6. The sEH Inhibitory Activities of Corn Gluten Peptides

Different enzymes were used to determine the sEH inhibitory activities of corn gluten hydrolysates. The results showed that corn gluten hydrolysates by Alcalase, papain, Flavourzyme, and pepsin and pancreatin all displayed good sEH inhibitory activities with a significant dose effect (*p* < 0.05) (Figure 5). The corn gluten hydrolysates by Alcalase had the lowest IC_50_ value of 1.07 ± 0.08 mg/mL. This is slightly lower than Flavourzyme, pepsin and pancreatin levels of 1.19 ± 0.24 and 1.46 ± 0.31 mg/mL, respectively, but no significance was observed (*p* > 0.05). However, the IC_50_ values of the corn gluten hydrolysates produced by Alcalase, Flavourzyme, and pepsin and pancreatin were all significantly lower than that of papain of 2.94 ± 0.26 mg/mL (*p* < 0.05). Different from Alcalase, papain, Flavourzyme, and pepsin and pancreatin, the corn gluten hydrolysates produced by Neutrase proteases had a low sEH inhibitory activity of less than 30% even at the concentration of 5 mg/mL (data not shown). Corn gluten hydrolysates by Alcalase, Flavourzyme, and pepsin and pancreatin were thought to be promising sources of natural sEH inhibitors.

What should be noted is that although the sEH inhibitory activity of corn gluten hydrolysate was not as high as most of the traditional urea and amide-containing synthetic chemicals and some natural compounds [7], the corn gluten hydrolysates by Alcalase, Flavourzyme, and pepsin and pancreatin could inhibit more than 90% of the sEH enzyme activity at the concentration of 5 mg/mL. Moreover, food protein hydrolysates generally have good water solubility and few adverse effects. They can be given safely at far higher concentrations compared with synthetic sEH inhibitors. This shows that food protein-derived sEH inhibitory peptides have potential as biological ingredients in functional foods or can be considered as a food with additional benefits.

The peptide sequences from the corn gluten hydrolysates by Alcalase were identified by nano-LC-MS/MS to further understand the sEH inhibitory activity of corn gluten hydrolysates. It was interesting to find that three of the above five peptides, WYW, YFW, and YFY, were successfully identified in the corn gluten hydrolysates by Alcalase (Figure 6). This may contribute to the superior sEH inhibitory activity of the corn gluten hydrolysates by Alcalase. Nevertheless, more sEH inhibitory peptides should be identified and verified in ongoing studies.

## 4. Conclusions

In this study, 8400 dipeptides and tripeptides were screened for potential sEH inhibitors using molecular docking. Finally, 30 peptides were selected and tested for sEH inhibitory activities. Peptides WEY, WWY, WYW, YFW, and YFY showed the highest activities, and all five synthetic peptides were competitive and non-competitive mixed inhibitors. Furthermore, sEH inhibitory activities were found in corn gluten hydrolysates produced by Alcalase, Papain, Flavourzyme, pepsin and pancreatin. In addition, three sEH inhibitory peptides WYW, YFW, and YFY were successfully identified in the corn gluten hydrolysates by Alcalase. Our results suggest that corn gluten peptides have potential as natural sEH inhibitors and corn gluten may be a prominent protein source for sEH inhibitory peptides.

## Figures and Tables

**Figure 1 foods-11-03695-f001:**
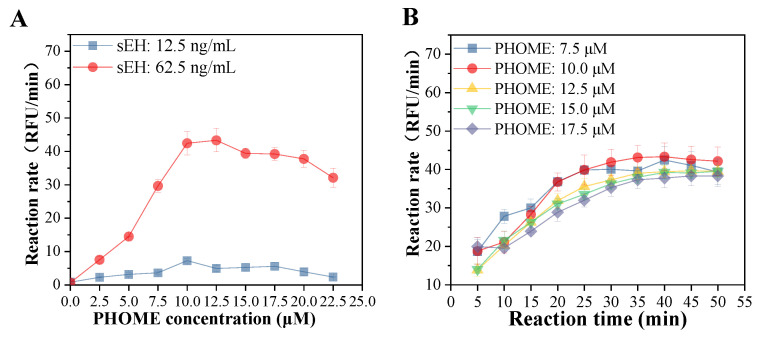
Effects of PHOME concentration (**A**) and reaction time (**B**) on the reaction rate of the hydrolase of sEH.

**Figure 2 foods-11-03695-f002:**
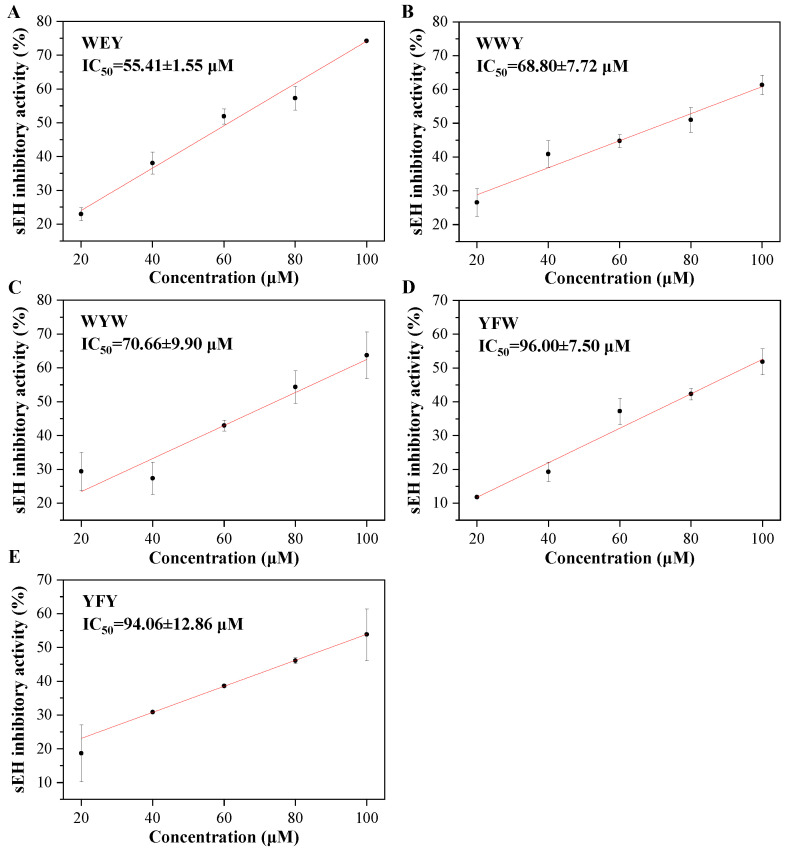
The sEH inhibitory activities of peptides WEY (**A**), WWY (**B**), WYW (**C**), YFW (**D**), and YFY (**E**).

**Figure 3 foods-11-03695-f003:**
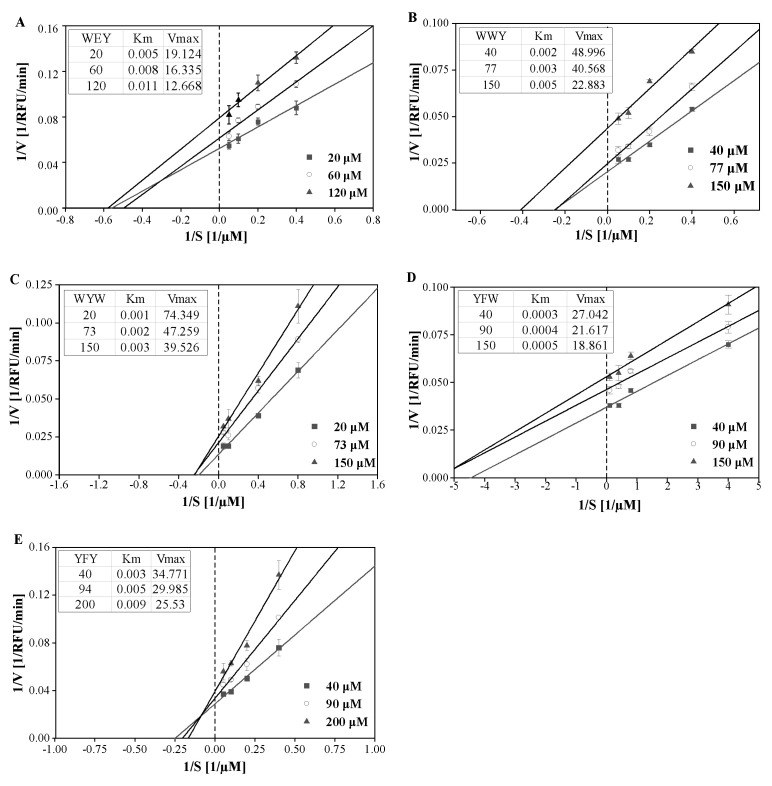
Lineweaver−Burk double-reciprocal plots of sEH inhibitory peptides WEY (**A**), WWY (**B**), WYW (**C**), YFW (**D**), and YFY (**E**).

**Figure 4 foods-11-03695-f004:**
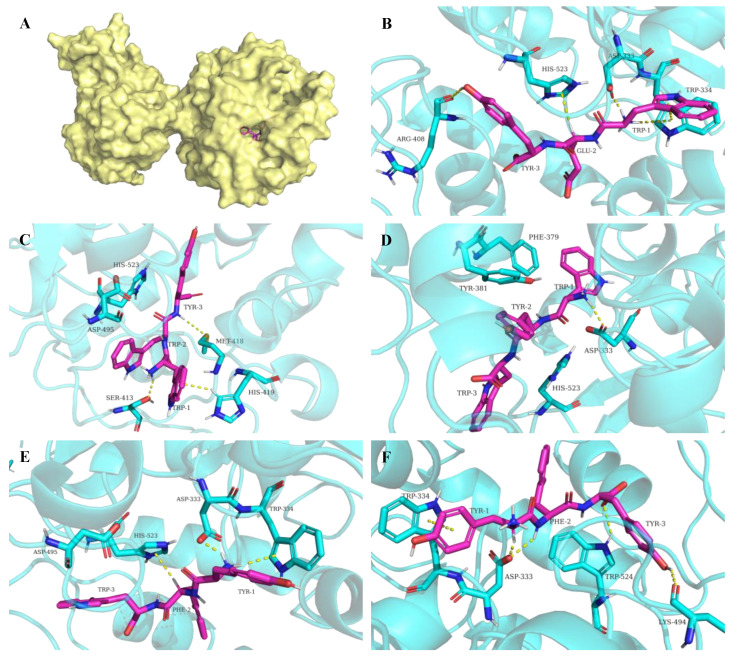
The conformation of sEH-peptide complex (**A**) and the interactions between sEH and peptides WEY (**B**), WWY (**C**), WYW (**D**), YFW (**E**), and YFY (**F**). The X-ray structure of human sEH (PDB code: 1ZD3) was obtained from the RCSB Protein Data Bank (https://www.rcsb.org, accessed on 10 October 2022) [21]. Molecular graphics were generated by PyMOL (http://www.pymol.org, accessed on 10 October 2022) [22].

**Figure 5 foods-11-03695-f005:**
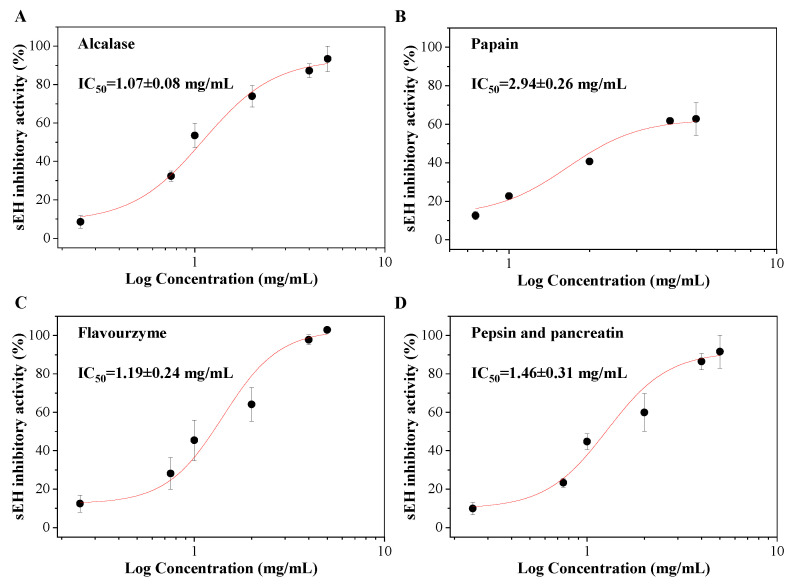
The sEH inhibitory activities of corn gluten hydrolysates by Alcalase (**A**), papain (**B**), Flavourzyme (**C**), and pepsin and pancreatin (**D**).

**Figure 6 foods-11-03695-f006:**
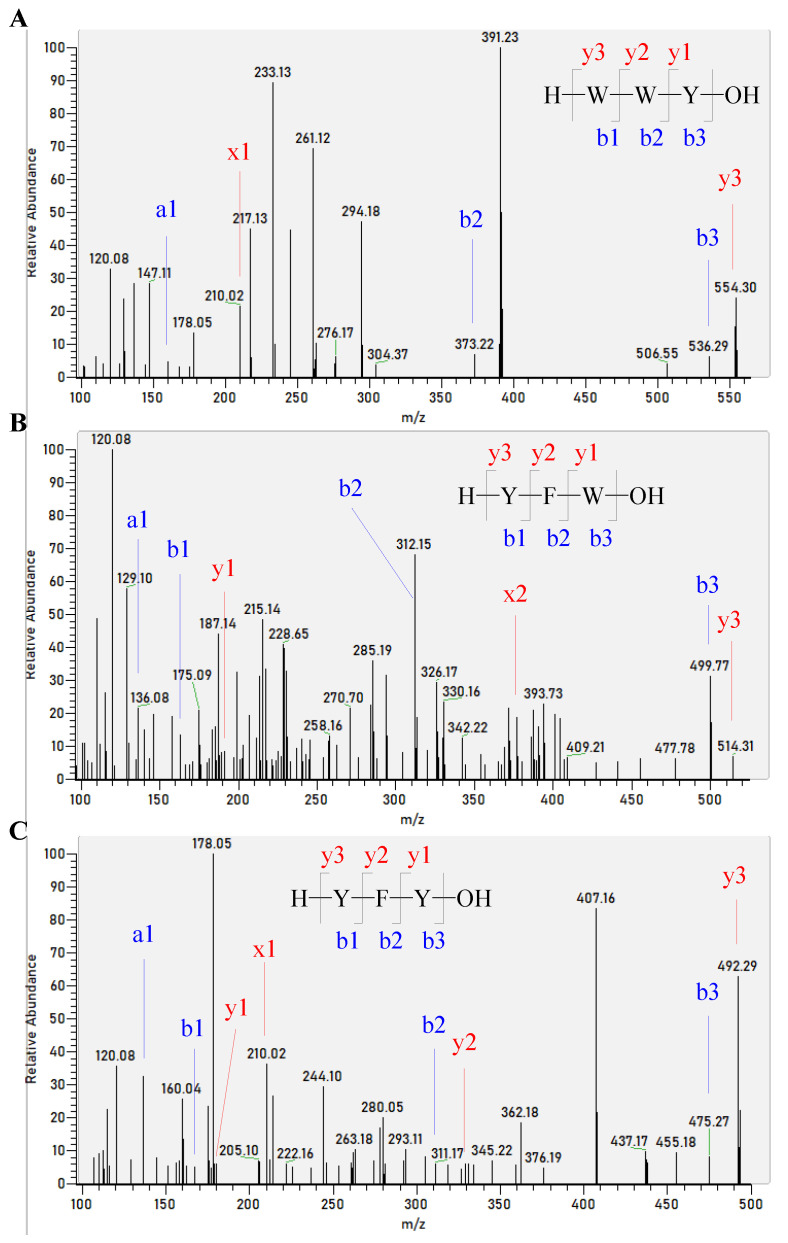
MS/MS spectra of singly charged ions with m/z at 554.30, 515.76, and 492.29 were determined to be the peptides WYW (**A**), YFW (**B**), and YFY (**C**), respectively.

**Table 1 foods-11-03695-t001:** The binding free energy of the top 20 tripeptides and 10 dipeptides and their sEH inhibitory activities.

Peptide Sequence	Binding Free Energy (kcal/mol)	sEH Inhibitory Activity at 100 µM (%)
KII	−11.0883	37.99 ± 8.23
WKW	−10.9831	31.13 ± 1.24
WQW	−10.8840	37.19 ± 6.14
WRW	−10.7671	38.30 ± 3.66
WLR	−10.5700	43.82 ± 1.73
YRW	−10.5607	47.58 ± 6.23
RWI	−10.5393	19.63 ± 10.26
FKW	−10.5095	36.29 ± 10.08
YMW	−10.4981	49.96 ± 5.86
WYW	−10.4858	60.30 ± 5.60
FRY	−10.4562	45.83 ± 14.14
WWY	−10.3874	55.15 ± 2.38
YFW	−10.3840	57.38 ± 7.62
YHW	−10.3641	32.06 ± 3.06
YFY	−10.3629	52.96 ± 3.09
WTR	−10.3607	26.05 ± 9.78
RYW	−10.3486	35.89 ± 2.83
YHY	−10.3050	41.43 ± 4.19
WEY	−10.3024	70.01 ± 6.25
WFW	−10.2877	36.24 ± 3.73
WR	−8.4954	31.65 ± 4.20
YW	−8.1424	32.08 ± 5.06
RV	−8.1371	46.76 ± 5.18
HR	−7.8854	29.71 ± 2.30
WW	−7.8543	39.88 ± 3.12
RW	−7.8351	19.84 ± 5.06
YR	−7.8328	27.32 ± 5.18
WF	−7.7965	43.98 ± 2.30
WY	−7.7922	29.16 ± 2.35
RY	−7.7852	22.75 ± 7.70

**Table 2 foods-11-03695-t002:** Summary of the interactions between peptides and sEH enzyme.

Peptide Sequence	Interactions	Site from Peptide			Site from She			Distance (Å)
WEY	hydrogen bond	Trp1	NH_3_^+^	H	Asp333	CO	O	2.0
		Tyr3	OH	H	Arg408	CO	O	2.3
	arene–cation stacking	Trp1	NH_3_^+^	H	Trp334	pyrrole ring		3.4
	arene–arene stacking	Trp1	pyrrole ring		Trp334	pyrrole ring		4.0
	arene–H stacking	Glu2	CH	H	His523	imidazole ring		3.0
WWY	hydrogen bond	Trp1	NH_3_^+^	H	Ser413	OH	O	2.2
		Tyr3	NH	H	Met418	S	S	3.0
	arene–H stacking	Trp1	benzene ring		His419	CH	H	2.8
WYW	hydrogen bond	Trp1	NH_3_^+^	H	Asp333	CO	O	2.1
YFW	hydrogen bond	Tyr1	NH_3_^+^	H	Asp333	OH	O	1.9
	arene–cation stacking	Tyr1	NH_3_^+^	H	Trp334	pyrrole ring		3.3
	arene–H stacking	Phe2	CH	H	His523	imidazole ring		2.8
YFY	hydrogen bond	Tyr1	CH	H	Asp333	OH	O	2.1
		Phe2	NH	H	Asp333	OH	O	2.2
		Tyr3	OH	H	Lys494	CO	O	2.2
		Tyr3	CO	O	Trp524	NH	H	2.5
	arene–arene stacking	Trp1	benzene ring		Trp334	pyrrole ring		3.6

## Data Availability

Data are contained within this article and Appendix A.

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
