# Peer review of "Screening and Identification of Novel Soluble Epoxide Hydrolase Inhibitors from Corn Gluten Peptides"

_foods, 2022, doi:10.3390/foods11223695_

Round 1

Reviewer 1 Report

The manuscript evaluates peptides from corn gluten with epoxide hydrolase inhibitory activity using an in silico and in vitro study.

First, the authors screened all the combinations of standard amino acids forming dipeptides (400) and tripeptides (800) to inhibit the soluble epoxide hydrolase (sEH) as a therapeutic target for many diseases using an in silico approach by molecular docking. Afterward, the best 30 candidate peptides were synthesized and their sEH inhibitory activity was evaluated in vitro (type inhibition and IC50 determination). Ultimately, IC50 of Corn gluten hydrolysates (alcalase, flavourzyme, papain, neutrase proteases) were determinate, and sEH inhibitory peptides were identified using nano LC-MS/MS.

I think that this study is a successful example of the use of in silico tools, to reduce the search space, in this case of sEH inhibitor peptides from corn gluten.

I only have some comments:

2.3. Virtual screening of potential sEH inhibitory peptides: The authors should describe the x,y,z coordinates for the docking and the steps follow for energy minimization of virtual peptides in each docking tool,

The authors declare that docking tool MOE showed the lowest RSMD in the redocking (Validation docking) of the native ligand NC4 into the binding site of sEH compared with AutoDock Vina and CDOCKER. Therefore, all dipeptides and tripeptides were docked to sEH using MOE. However, in lines 196 to 198, the authors declared the following: “The root means square deviation (RMSD) between the redocked pose and the native pose in MOE, AutoDock Vina, and CDOCKER were calculated to be 1.939, 1.062, and 2.445 Å, respectively.” Shows that RSMD obtained by AutoDock Vina is the lowest (1.062 Å vs 1.939 Å; 2.445 Å). Why not use AutoDock Vina to dock the dipeptides and tripeptides to sEH? please clarify.

In Table 1, the authors indicate that “data marked by different letters were significantly different (P<0.05)”. But I can’t find different letters in the table.

In Table 1, the tripeptide KII shows the lower binding free energy (-11.0883 kcal/mol) but does not produce the highest sEH inhibitory activity at 100 μM (37.99 vs 70.01). How can the authors explain these results? Please discuss this.

The authors state in the introduction section the undesirables’ pharmacokinetic properties of sEH inhibitors from traditional Chinese herbs and the importance of discovering possible sEH inhibitors from common nutritional components of foods. I think that including the evaluations of pharmacokinetics properties of the sEH inhibitory peptides from corn gluten will complement the manuscript results interpretation and scope. I recommend use ADMETlab 2.0, an online platform for this purpose (ADMETlab 2.0: An Integrated Online Platform for Accurate and Comprehensive Predictions of ADMET Properties; Doi: 10.1093/nar/gkab255).

This is a research published in the prestigious journal “Foods” of MDPI as an example:

Arámburo-Gálvez, J. G., Arvizu-Flores, A. A., Cárdenas-Torres, F. I., Cabrera-Chávez, F., Ramírez-Torres, G. I., Flores-Mendoza, L. K., ... & Ontiveros, N. (2022). Prediction of ACE-I Inhibitory Peptides Derived from Chickpea (Cicer arietinum L.): In Silico Assessments Using Simulated Enzymatic Hydrolysis, Molecular Docking and ADMET Evaluation. Foods, 11(11), 1576. Doi: 10.3390/foods11111576

Author Response

Reviewer 1#:

The manuscript evaluates peptides from corn gluten with epoxide hydrolase inhibitory activity using an in silico and in vitro study.

First, the authors screened all the combinations of standard amino acids forming dipeptides (400) and tripeptides (800) to inhibit the soluble epoxide hydrolase (sEH) as a therapeutic target for many diseases using an in silico approach by molecular docking. Afterward, the best 30 candidate peptides were synthesized and their sEH inhibitory activity was evaluated in vitro (type inhibition and IC50 determination). Ultimately, IC50 of Corn gluten hydrolysates (alcalase, flavourzyme, papain, neutrase proteases) were determinate, and sEH inhibitory peptides were identified using nano LC-MS/MS.

I think that this study is a successful example of the use of in silico tools, to reduce the search space, in this case of sEH inhibitor peptides from corn gluten.

Response: Thank you very much for your review. Your comments are all valuable and helpful for improving our manuscript. According to your comments, modifications have made to our manuscript.

I only have some comments:

2.3. Virtual screening of potential sEH inhibitory peptides: The authors should describe the x,y,z coordinates for the docking and the steps follow for energy minimization of virtual peptides in each docking tool.

Response: Thank you for your suggestion. The information of the x,y,z coordinates of the binding site for the docking and the detailed steps for minimizing energy in each docking tool had been added in the manuscript.

The authors declare that docking tool MOE showed the lowest RSMD in the redocking (Validation docking) of the native ligand NC4 into the binding site of sEH compared with AutoDock Vina and CDOCKER. Therefore, all dipeptides and tripeptides were docked to sEH using MOE. However, in lines 196 to 198, the authors declared the following: “The root means square deviation (RMSD) between the redocked pose and the native pose in MOE, AutoDock Vina, and CDOCKER were calculated to be 1.939, 1.062, and 2.445 Å, respectively.” Shows that RSMD obtained by AutoDock Vina is the lowest (1.062 Å vs 1.939 Å; 2.445 Å). Why not use AutoDock Vina to dock the dipeptides and tripeptides to sEH? please clarify.

Response: We feel sorry for our carelessness. A mistake had been made on the RMSD results. In fact, the RMSD values between the redocked pose and the native pose in MOE, AutoDock Vina, and CDOCKER were calculated to be 1.062, 1.939 and 2.445 Å, respectively. The RMSD values had been revised in the manuscript.

In Table 1, the authors indicate that “data marked by different letters were significantly different (P<0.05)”. But I can’t find different letters in the table.

Response: Thank you for your comments. The sentence “The data marked by different letters were significantly different (P<0.05)” had been deleted.

* In Table 1, the tripeptide KII shows the lower binding free energy (-11.0883 kcal/mol) but does not produce the highest sEH inhibitory activity at 100 μM (37.99 vs 70.01). How can the authors explain these results? Please discuss this.

Response: Thank you for your comments. As you mentioned, the lower binding free energy usually represents the more stable conformation of the ligand-receptor complex and stronger biological activity. Therefore, the peptides with the lower binding free energy to sEH are predicted to have the potential to show higher activities in theory. This prediction method helps to shorten the time and cost of the screen of bioactive compounds. However, it is noted that, in some times, the values of the binding free energy are not always absolutely correlated with the real biological activity. This is why the top 20 tripeptides and 10 dipeptides were selected to better screen the potential sEH inhibitors in our manuscript.

The authors state in the introduction section the undesirables’ pharmacokinetic properties of sEH inhibitors from traditional Chinese herbs and the importance of discovering possible sEH inhibitors from common nutritional components of foods. I think that including the evaluations of pharmacokinetics properties of the sEH inhibitory peptides from corn gluten will complement the manuscript results interpretation and scope. I recommend use ADMET lab 2.0, an online platform for this purpose. (ADMETlab 2.0: An Integrated Online Platform for Accurate and Comprehensive Predictions of ADMET Properties; Doi: 10.1093/nar/gkab255). This is a research published in the prestigious journal “Foods” of MDPI as an example: Ará mburo-Gá lvez, J. G., Arvizu-Flores, A. A., Cá rdenas-Torres, F. I., Cabrera-Chá vez, F., Ramí rez-Torres, G. I., Flores-Mendoza, L. K., ... & Ontiveros, N. (2022). Prediction of ACE-I Inhibitory Peptides Derived from Chickpea (Cicer arietinum L.): In Silico Assessments Using Simulated Enzymatic Hydrolysis, Molecular Docking and ADMET Evaluation. Foods, 11(11), 1576. Doi: 10.3390/foods11111576

Response: Thank you for your suggestions. We tried to predict the pharmacokinetic properties of the sEH inhibitory peptides using ADMETlab 2.0. However, it was found that most of the sEH inhibitory peptides in our manuscript did not work well due to their high molecular weight over than 500 Da. Therefore, it is not suitable to predict the pharmacokinetic properties of the sEH inhibitory peptides using ADMETlab 2.0 as well as some small compounds. Alternatively, the physicochemical properties including iso-electric point (pI), net charge at neutral pH, and water solubility of identified peptides were analyzed by the peptide property calculator available at http://www.innovagen.com/proteomics-tools. The toxicity of peptides was predicted using ToxinPred available at https://webs.iiitd.edu.in/raghava/toxinpred/multi_submit.php. The above results had been summarized in Table S1, which had been added in the manuscript and supplementary data. Moreover, the absorption in the intestine, bioavailability and half-time in animals and humans will be determined in the future.

In addition, some descriptions and discussion on the results had also been added in the manuscript as follows: “The physicochemical properties of the identified peptides were analyzed by the peptide property calculator (Table S1). The peptide WEY had a negative charge at pH 7 and good solubility. The other peptides showed electroneutral and poor water solubility. All peptides were predicted without toxicity. These results provided basis for the further analysis of the pharmacokinetic properties including the absorption in the intestine, bi-oavailability and half-time in animals and humans in the future.”.

Table S1: Physicochemical properties and toxicity of sEH inhibitory peptides.

No.

Sequence

MW (Da)

pI

Net charge at pH 7

Water solubility

Toxicity

1

WEY

496.52

0.95

-1

good

non-toxic

2

WYW

553.61

3.44

0

poor

non-toxic

3

YFW

514.57

3.31

0

poor

non-toxic

4

WWY

553.61

3.51

0

poor

non-toxic

5

YFY

491.53

3.37

0

poor

non-toxic

Reviewer 2 Report

Review of Screening and identification of novel soluble epoxide hydrolase inhibitors from corn gluten proteins.  Jiamin Dang et al.  No 4, 2022. Corresponding author is Liying Wang <[email protected].cn>

The abstract and introduction are well written and clear.  The authors make an excellent case for looking at natural products in general and peptides in particular. Over all the paper is well documented with appropriate references, the figures are clearly explained and relevant and the discussion matches the data presented.  The idea of using protein hydrolysates in this was to control inflammation by inhibition of the soluble epoxide hydrolase enzyme is novel and well carried out.  Acceptance of the paper for publication is recommended.  Minor points for the authors to consider are listed below.

Minor points.

In our previous study, it was found that corn gluten-derived peptides by 55
Alcalase exhibited prominent free radicals and intracellular reactive oxygen species 56
scavenging capacities [15,16].  Better to say peptides generated by gluten hydrolysis by the enzyme Alcalase.

In this study, the potential sEH inhibitory capacity of peptides was first explored by 61
docking the peptides into the active site of sEH….computational docking of the peptides into…

It is great authors used a standard sEHI for comparison. AUDA is adequate but a very old compound.  The more commonly used inhibitor it TPPU also available from Cayman.  It is more structurally related to the compounds in current human trials.  The authors should go back to original papers on assays for the sEH and inhibitors.  Cayman’s instructions can be miss leading and leave out cautions for use of the assay.  It is also good to reference original papers on the production of reagents like AUDA.

After cooled down (Alcalase: 50 °C; Flavourzyme: 50 86
°C; papain: 55 °C; Neutrase proteases: 45 °C),  after cooling down

The autodoc work is well described.

Adding 130 ul of sEH is the wrong unit.  That refers one assumes to a Caymen product.  One could refer to the enzyme in nanomoles, micrograms, catalytic activity but not microliters.  50 ul of PHOME is also an incorrect unit.  Cayman kits come and go. One assumes that the peptide extract does not contain esterases or glutathione S-transferases which cause high background.  Later expressing sEH as ‘sEH enzyme concentration of 62.5 ng/mL was used.’ Is the proper unit.  Same with PHOME.

The authors give 4 significant figures for their inhibition results while their data support two.  Thus showing three figures for inhibition is more than adequate.

The authors should note that there are synthetic sEHI reported working with Ki’s in the low picomolar range.  These slow tight binding inhibitors violate the assumptions of Michaels Menton and thus Lineweaver-Burk Plots are not adequate.  This can just be mentioned – because the kinetics done by the authors seem very adequate. 

The NH of the urea 271
or amide group of many sEH inhibitors can act as a hydrogen bond donor and form a 272
hydrogen bond with the oxygen atom of Asp333. Therefore, various urea and am- 273
ide-containing compounds have been demonstrated as competitive inhibitors for sEH [6]  The urea and amide are thought to form a hydrogen bond stabilized salt bridge in the catalytic site.  The authors might consider this in their peptide design.

The author’s statement is well justified by their data. Possibly they should bring up the massive difference between synthetic sEHI potency and the peptides.  However, the peptides could be given safely at far higher concentrations.    What should be noted was that although the sEH inhibitory activity of corn gluten 327
hydrolysate was not as high as most of the traditional urea and amide-containing syn- 328
thesized chemicals and some natural compounds [7], the corn gluten hydrolysates by 329
Alcalase, Flavourzyme, and pepsin and pancreatin could inhibit more than 90% of the 330
sEH enzyme activity at the concentration of 5 mg/mL. Moreover, food protein hydroly- 331
sates generally have good water solubility and little adverse effects and can be used as 332
functional ingredients in foods at a high dosage.

Line 340     and verified in on going studies.

Just as suggestions the authors could bring up the following points.

‘The speed and accuracy of structure predictions and docking models have improved dramatically in just the last two years.’

‘Although the peptides are far less potent than known synthetic inhibitors of the sEH, these peptides are food products and can be considered as a food with additional benefits.’

‘From a more basic standpoint the structure activity relationships among these peptides can be used to further refine structural models of the sEH improving the power of future computational approaches.’

‘Peptides vary dramatically in their absorption and pharmacokinetics however, inhibitors of the sEH have been found to improve gut barrier function and reduce inflammation of the intestinal epithelium.  Thus, the local activity of these compounds in the gastrointestinal tract could reduce disease symptoms locally in the gastrointestinal tract and systemically by reducing the loss of barrier function of the intestine.’ 

Author Response

Reviewer 2#:

The abstract and introduction are well written and clear. The authors make an excellent case for looking at natural products in general and peptides in particular. Over all the paper is well documented with appropriate references, the figures are clearly explained and relevant and the discussion matches the data presented. The idea of using protein hydrolysates in this was to control inflammation by inhibition of the soluble epoxide hydrolase enzyme is novel and well carried out. Acceptance of the paper for publication is recommended. Minor points for the authors to consider are listed below.

Response: Thank you very much for your review on our manuscript and your encouraging comments. We also appreciate your clear and detailed comments and hope that the revision has fully addressed your concerns.

In our previous study, it was found that corn gluten-derived peptides by Alcalase exhibited prominent free radicals and intracellular reactive oxygen species scavenging capacities [15,16]. Better to say peptides generated by gluten hydrolysis by the enzyme Alcalase.

Response: We sincerely thank your careful reading. As suggested by you, this sentence had been revised as “In our previous study, it was found peptides generated from corn gluten hydrolysis by the enzyme Alcalase”.

In this study, the potential sEH inhibitory capacity of peptides was first explored by docking the peptides into the active site of sEH… computational docking of the peptides into…

Response: Thanks for your suggestion. This sentence had been revised as “In this study, the potential sEH inhibitory capacity of peptides was first explored by computational docking of the peptides into the active site of sEH”.

It is great authors used a standard sEHI for comparison. AUDA is adequate but a very old compound. The more commonly used inhibitor it TPPU also available from Cayman. It is more structurally related to the compounds in current human trials. The authors should go back to original papers on assays for the sEH and inhibitors. Cayman’s instructions can be miss leading and leave out cautions for use of the assay. It is also good to reference original papers on the production of reagents like AUDA.

Response: Thank you for your suggestion. AUDA is a well-known and recognized sEH inhibitor [1,2]. TPPU is also a commonly used inhibitor as you mentioned. We appreciate your suggestion to return to the original paper on assays for the sEH and inhibitors and will carefully check the instructions. In our next work, TPPU will be selected as the positive control.

References:

[1] Wang, H. L.; Chen ,J. W.; Yang, S. H.; Lo, Y. C.; Pan, H. C.; Liang, Y. W.; Wang, C. F.; Yang, Y.; Kuo, Y. T.; Lin, Y. C.; Chou, C. Y.; Lin, S. H.; Chen, Y. Y. Multimodal Optical Imaging to Investigate Spatiotemporal Changes in Cerebrovascular Function in AUDA Treatment of Acute Ischemic Stroke. Front Cell Neurosci. 2021, 15, 655305. https://doi.org/10.3389/fncel.2021.655305.

[2] Dong, X. W.; Jia, Y. L.; Ge, L. T.; Jiang, B.; Jiang, J. X.; Shen, J.; Jin, Y. C.; Guan, Y.; Sun, Y.; Xie, Q. M. Soluble epoxide hydrolase inhibitor AUDA decreases bleomycin-induced pulmonary toxicity in mice by inhibiting the p38/Smad3 pathways. Toxicology. 2017, 389, 31-41. https://doi.org/10.1016/j.tox.2017.07.002.

After cooled down (Alcalase: 50 ℃; Flavourzyme: 50 °C; papain: 55 °C; Neutrase proteases: 45 °C), after cooling down.

Response: The word “cooled” had been changed to “cooling”.

The autodoc work is well described. Adding 130 µl of sEH is the wrong unit. That refers one assumes to a Caymen product. One could refer to the enzyme in nanomoles, micrograms, catalytic activity but not microliters. 50 µl of PHOME is also an incorrect unit. Cayman kits come and go. One assumes that the peptide extract does not contain esterases or glutathione S-transferases which cause high background. Later expressing sEH as ‘sEH enzyme concentration of 62.5 ng/mL was used.’ Is the proper unit. Same with PHOME.

Response: Thank you for your comments. The description on sEH and PHOME had been revised in the manuscript.

The authors give 4 significant figures for their inhibition results while their data support two. Thus showing three figures for inhibition is more than adequate.

Response: Thank you for your comments. We think that more figures will help to better understand the results in the manuscript.

The authors should note that there are synthetic sEHI reported working with Ki’s in the low picomolar range. These slow tight binding inhibitors violate the assumptions of Michaels Menton and thus Lineweaver-Burk Plots are not adequate. This can just be mentioned – because the kinetics done by the authors seem very adequate.

Response: Thank you for your suggestions. The discussion on the limitations of the Lineweaver Burk double recursive plots had been added in the manuscript as follows: “What should be noted was that the adequate of the Lineweaver-Burk double-reciprocal plots in this study may be due to the low binding affinity of peptides to sEH.”.

The NH of the urea or amide group of many sEH inhibitors can act as a hydrogen bond donor and form a hydrogen bond with the oxygen atom of Asp333. Therefore, various urea and amide-containing compounds have been demonstrated as competitive inhibitors for sEH [6]. The urea and amide are thought to form a hydrogen bond stabilized salt bridge in the catalytic site. The authors might consider this in their peptide design.

Response: Thank you for your suggestions. It is known that many sEH inhibitors have urea and amide groups. The peptide bound bond in peptides is one of the amide groups. Therefore, the sEH inhibitory properties of peptides were investigated in the present study. Molecular docking analysis revealed that the sEH inhibitory peptides were predicted to mainly form hydrogen bond and arene-cation stacking, arene-arene stacking, and arene-H stacking interactions with the residues of the active site of sEH. The hydrogen bond stabilized salt bridge interactions between peptide and sEH were not observed in the present study. However, it will be considered in our next work.

The author’s statement is well justified by their data. Possibly they should bring up the massive difference between synthetic sEHI potency and the peptides. However, the peptides could be given safely at far higher concentrations. What should be noted was that although the sEH inhibitory activity of corn gluten hydrolysate was not as high as most of the traditional urea and amide-containing synthesized chemicals and some natural compounds [7], the corn gluten hydrolysates by Alcalase, Flavourzyme, and pepsin and pancreatin could inhibit more than 90% of the sEH enzyme activity at the concentration of 5 mg/mL. Moreover, food protein hydroly sates generally have good water solubility and little adverse effects and can be used as functional ingredients in foods at a high dosage.

Response: Yes, as you mentioned, the food protein-derived sEH inhibitory peptides have many advantages compared with synthetic sEH inhibitors. The related discussion had been revised as “They can be given safely at far higher concentrations compared with synthetic sEH inhibitors. It shows that food protein-derived sEH inhibitory peptides have the potential and can be used as functional biological ingredients in functional foods or can be considered as a food with additional benefits.” in the manuscript.

Line 340  and verified in on going studies.

Response: It had been revised.

Just as suggestions the authors could bring up the following points.

The speed and accuracy of structure predictions and docking models have improved dramatically in just the last two years.’

Although the peptides are far less potent than known synthetic inhibitors of the sEH, these peptides are food products and can be considered as a food with additional benefits.’

From a more basic standpoint the structure activity relationships among these

peptides can be used to further refine structural models of the sEH improving the power of future computational approaches.’

Peptides vary dramatically in their absorption and pharmacokinetics however, inhibitors of the sEH have been found to improve gut barrier function and reduce inflammation of the intestinal epithelium. Thus, the local activity of these compounds in the gastrointestinal tract could reduce disease symptoms locally in the gastrointestinal tract and systemically by reducing the loss of barrier function of the intestine.’

Response: Thank you very much for sharing these beautiful sentences, which will play an important role in enriching the content of the manuscript. We have applied these helpful sentences to the appropriate position in the revised manuscript.